# Mechanical Properties and Microstructural Evolution of TiNi-Based Intermetallic Alloy with Nb Addition

**DOI:** 10.3390/ma15093124

**Published:** 2022-04-26

**Authors:** Hsin-Feng Yang, Tao-Hsing Chen, Ying-Ying Syu

**Affiliations:** Department of Mechanical Engineering, National Kaohsiung University of Science and Technology, Kaohsiung 80778, Taiwan; yanghf@nkust.edu.tw (H.-F.Y.); stars0619@gmail.com (Y.-Y.S.)

**Keywords:** TiNi-based intermetallic alloy, universal testing machine, split-Hopkinson pressure bar, mechanical properties

## Abstract

TiNi intermetallic alloys were prepared with 2, 4 and 6 at.% niobium (Nb) addition. The mechanical properties and microstructures of the alloys were investigated under both static (1 × 10^−1^ to 1 × 10^−3^ s^−1^) and dynamic (4 × 10^3^ to 6 × 10^3^ s^−1^) loading conditions. The intermetallic alloy structures and surface morphologies of the alloys were examined by X-ray diffraction (XRD) and scanning electron microscopy (SEM), respectively. In addition, the fracture morphologies were observed by optical microscopy (OM). It was shown that the addition of 2 to 4 at.% Nb increased the strength of the TiNi alloy. However, as the level of Nb addition was further increased to 6 at.%, a significant reduction in strength occurred. For a constant Nb addition, the plastic flow stress and strain rate sensitivity increased with increasing strain rate under both loading conditions (static and dynamic). The XRD and SEM results showed that the original surface morphologies were composed primarily of dendritic structures and fine β-Nb + TiNi eutectic systems. Moreover, the OM results showed that the alloys underwent a transition from a brittle fracture mode to a ductile fracture mode as the level of Nb addition increased.

## 1. Introduction

Intermetallic compounds (IMCs) consist of two or more metallic or metalloid elements with specific atomic ratios and have an ordered crystalline structure. Such alloys have an attractive combination of physical and mechanical properties, including a high melting point, light weight, high strength, and good oxidation and creep resistance. As a result, they are regarded as promising materials for a wide range of applications in the aerospace, automobile, and nuclear energy fields [1,2,3]. Many IMCs have been developed in recent decades, including FeAl, Ni_3_Al, TiAl and TiNi [4,5,6,7,8,9,10]. Among these alloys, TiNi intermetallic alloys have been used for many practical applications in biomedical and engineering fields due to their excellent performance, such as biocompatibility, anti-corrosion and high mechanical properties and shape memory effect. However, the TiNi alloy waas first used for shape memory due to good transform hysteresis [11,12,13]. TiNi alloy also has a hardness comparable to that of tool steel, and good superelasticity. So, it is one of the most promising alloy systems for bearings, automobiles and related applications [14]. TiNi-based alloys have also found widespread application in fields such as aerospace engineering, intelligent control, and medical implants [15,16,17]. As with all IMCs, TiNi has good mechanical strength at elevated temperatures; however, it exhibits relatively brittle behavior under ambient temperature conditions, and thus has restricted applicability at lower working temperatures.

The microstructure of TiNi alloy is B2 high temperature matrix phase. The microstructure has super-elasticity and shape memory; this makes TiNi alloys difficult to be machined and to be highly resistant to deformation. These factors are due to diffusionless martensitic transformation. Over recent decades, many studies have been conducted on the effect of using ternary alloying elements for the improvement of mechanical properties or grain refinement treatment [18,19]. Many studies have shown that the properties of IMCs can be tuned through the addition of carefully chosen doping elements. For example, the addition of hafnium (Hf) reduces the solid solution temperature and is hence advantageous in lowering manufacturing costs at an industrial scale [20,21]. Similarly, the addition of Al induces a grain refinement effect, which is beneficial in enhancing IMC hardness [22,23]. The addition of Nb as a doping element has attracted particular attention due to its effects in enhancing the yield strength through the formation of ultrafine grains, good corrosion resistance at high temperature, and high dislocation density [24,25,26]. Co-addition is also beneficial in improving the solid solution strengthening effect and magnetic hysteresis, thereby improving the thermal coupling performance and oxidation resistance [27]. During the forming process and application, the TiNi alloy will suffer differential strain rate deformation, so it is necessary to investigate the strain rate effect on the TiNi alloy. Intermetallic alloys can be fabricated through various methods, including casting, powder metallurgy, self-propagation high-temperature synthesis (SHS), directional solidification, selective laser melting, and electron beam melting [28,29,30]. However, the vacuum arc melting method is easy to operate and a low-cost method. This study used this method to fabricate the experimental specimens.

Some studies have reported that the addition percentage of Nb is high, but will enhance the high yielding strength due to the β-Nb phase composite mechanism [31,32]. However, high yielding strength was also shown to be associated with decreased ductility. It is important to achieve good mechanical properties including both strength and ductility. Accordingly, the present study prepared TiNi alloys with 2, 4 and 6 at.% Nb addition and investigated their mechanical properties and microstructural evolution over a wide strain rate range of 10^−3^ s^−1^ to 6 × 10^3^ s^−1^. The effects of the precipitation phase on the mechanical properties of the TiNi-Nb alloys were examined by X-ray diffraction (XRD) and scanning electron microscopy (SEM). Finally, the fracture mechanisms of the various alloys were investigated by observing fracture surfaces using an SEM facility.

## 2. Experimental Procedures

In the present study, TiNi-Nb alloys were prepared in a vacuum arc melting furnace under argon gas. High-purity (99.99%) Ti, Ni and Nb powders were purchased from the Golden Optoelectronic Company (Taipei, Taiwan). The TiNi alloys were prepared using a constant Al content of 44 at.%, Nb contents of 2, 4 or 6 at.%, and a balance of Ti. The TiNi_44_Nb_2_, TiNi_44_Nb_4_ and TiNi_44_Nb_6_ ingots were melted at temperatures between 1573 K and 1673 K for 6 h, allowed to cool to room temperature in the furnace, and then reheated once again. To ensure compositional homogeneity, each ingot was remelted five times. Furthermore, to ensure homogenization, the homogenization process was performed keeping the alloys at 1060 °C for 20 h in a vacuum furnace. The as-cast ingots were machined into cylindrical bars with a diameter of 5 mm. and the bars were then cut into test specimens with a length of 5 mm using a low-speed cutter. The phase compositions and microstructures of the three alloys were examined via XRD analysis and SEM (JEM7001, JEOL Ltd., Akishima, Japan). The mechanical characteristics of each specimen were then investigated under room temperature conditions using quasi-static and dynamic tests. In the quasi-static tests, the specimens were deformed at strain rates of 10^−3^ s^−1^, 10^−2^ s^−1^ and 10^−1^ s^−1^, respectively, using a material testing system (MTS Landmark; Sinodynamics Enterprise Co., Ltd., Taipei, Taiwan). The dynamic tests were performed under lubricated conditions using a compression split-Hopkinson pressure bar (SHPB) at strain rates of 4000 s^−1^, 5000 s^−1^ and 6000 s^−1^, respectively. Finally, the fracture surfaces of the specimens were observed by SEM.

## 3. Results and Discussion

### 3.1. XRD Structural Analysis

XRD analyses were performed to identify the intermetallic alloy structures of the as-cast TiNi_44_Nb_2_, TiNi_44_Nb_4_ and TiNi_44_Nb_6_ alloys. The XRD pattern shown in Figure 1a for the TiNi_44_Nb_2_ alloy contained prominent peaks corresponding to the TiNi (B2) phase in the (110), (200) and (211) directions, together with several peaks corresponding to Ti_2_Ni. As the Nb addition increased to 4 at.%, the intensity of the Ti_2_Ni peaks decreased substantially with the peak at 42°–50° disappearing completely. Finally, for 6 at.% Nb addition, a soft β-Nb phase with a BCC crystal structure was newly formed. In other words, the addition of a larger quantity of Nb produced a mixed TiNi (B2) + βNb phase structure through a eutectic reaction.

### 3.2. Surface Morphology Analysis

In order to observe microstructure, all the tested specimens were cold-coated with polyester resin, and then polished. Then, the specimens were etched in chemical solution with a volume ratio of HF:HNO_3_:H_2_O = 1:1:7 for 20 s and their microstructures and chemical composition were determined with SEM and EDX systems. Figure 2a–c present SEM surface morphology images of the TiNi_44_Nb_2_, TiNi_44_Nb_4_ and TiNi_44_Nb_6_ alloys. It can be seen that all the alloys comprise a dendritic TiNi (B2) substrate with a black Ti_2_Ni intermetallic phase. As the level of Nb addition increased, the alloy structure became more refined, and the dendritic structure became denser. As indicated by the XRD analysis results, the TiNi_44_Nb_6_ alloy was composed of _a_ dendritic TiNi (B2) matrix together with eutectic structures around the grain boundaries. Figure 2d,e present the EDX analysis for Figure 2a–c, respectively. From the EDX analysis, the microstructure containing Ti, Ni and Nb elements can be seen. The primary and secondary dendrites for TiNiNb_2_ were about 40 μm and 5 μm in size. The primary and secondary dendrites for TiNiNb_4_ were about 30 μm and 10 μm in size. The primary dendrites for TiNiNb_6_ were less than 10 μm and almost disappeared, while the secondary dendrites for TiNiNb_6_ were about 12 μm and became discontinuous. The precipitate β-Nb in the matrix can also be seen (Figure 2g is the large magnification figure for Figure 2c). In the literature, it is suggested that as the Nb content increases the temperature of the phase transformation and related mechanical properties will increase [33,34].

### 3.3. Mechanical Properties

Figure 3, Figure 4 and Figure 5 show the mechanical response of the TiNi_44_Nb_2_, TiNi_44_Nb_4_ and TiNi_44_Nb_6_ alloys, respectively, under quasi-static and dynamic compressive loads. In general, the results show that for Nb additions of less than 4 at.%, an increasing Nb content led to a higher failure stress and failure strain. In other words, the interstitial insertion of the Nb atoms prompted a distortion of the TiNi lattice, which increased the lattice strain and impeded the movement of dislocations, thereby resulting in a solid solution strengthening effect. However, as the level of Nb addition was further increased to 6 at.%, the strength decreased and the ductility increased due to the formation of a soft β-Nb phase, as shown in Figure 1 and Figure 2.

### 3.4. Strain Rate Sensitivity Effect

Figure 6 and Figure 7 show the variation in the flow stress with the strain rate for the three alloys as a function of the strain under static and dynamic loading conditions, respectively. It can be seen that, for both deformation regimes, the strain rate and Nb content had a significant effect on the plastic flow stress. Moreover, for a constant strain rate, the flow stress increased with increasing strain. Among all the alloys, the TiNi_44_Nb_4_ alloy exhibited the maximum plastic flow stress for all values of the strain and strain rate. The effect of the strain rate and Nb content on the mechanical response of the three alloys can be evaluated via the following strain rate sensitivity (*β*) [35]:$$\beta =\frac{\sigma_{2}-\sigma_{1}}{\ln\left({\dot{\varepsilon }}_{2}/{\dot{\varepsilon }}_{1}\right)}$$

Figure 8 and Figure 9 show the variation in the strain rate sensitivity exponent with the true strain as a function of the strain rate under quasi-static and dynamic loading conditions, respectively. For both deformation regimes, and all values of the strain rate, the strain rate sensitivity increased with increasing strain. In other words, the strain rate increased with increasing strain. The maximum strain rate sensitivity occurred in the TiNi_44_Nb_4_ alloy, irrespective of the loading condition (quasi-static or dynamic). Moreover, the strain rate sensitivity reduced significantly in the TiNi_44_Nb_6_ specimen with 6 at.% Nb addition. It is speculated that the reduced strength and enhanced ductility of the TiNi_44_Nb_6_ alloy was due to the softening effects of the β-Nb phase formed at the grain boundaries, as shown in Figure 2c.

### 3.5. Fracture Surface Morphology Analysis

As shown in the stress-strain curves in Section 3.3, all the specimens failed under the considered quasi-static and dynamic strain rates. To investigate the failure mode for each sample, the facture surfaces were examined by SEM. Figure 10a,b show the fracture surface of the TiNi_44_Nb_2_ alloy under quasi-static and dynamic strain rates of 10^−1^ s^−1^ and 5 × 10^3^ s^−1^, respectively. For both strain rates, the fracture surface contained multiple fracture planes and numerous cracks at the grain boundary, which suggests that the onset of specimen failure occurred at the grain boundaries. Overall, the results show that the failure mode was one of intergranular fracture (i.e., brittle failure). For the TiNi_44_Nb_4_ alloy, both facture surfaces contained intergranular fractures and granular equiaxed dimples. In other words, the fracture surfaces showed evidence of a mixed brittle-ductile failure mode (see Figure 11). For the TiNi_44_Nb_6_ alloy, the addition of a greater amount of Nb reduced the number of intergranular fracture features and increased the density (i.e., reduced the size) of the equiaxed dimples (see Figure 12). Overall, the OM images shown in Figure 10, Figure 11 and Figure 12 indicate that the addition of 2–4 at/% Nb resulted in a grain refinement effect, which prompted solid solution strengthening of the alloy with corresponding improvement in the mechanical performance. However, for a higher level of Nb addition (6 at.%), a soft β-Nb phase was formed, which reduced the strength of the alloy and increased its ductility.

## 4. Conclusions

TiNi intermetallic alloys were prepared with 2, 4 and 6 at.% niobium (Nb) addition. The mechanical properties and microstructures of the alloys were investigated under both static and dynamic loading conditions. The experimental results support the following main conclusions.

The TiNi_44_Nb_2_ and TiNi_44_Nb_4_ alloys were composed principally of TiNi and Ti_2_Ni phase. However, for a higher Nb addition of 6 at.%, the microstructure contained a mixture of TiNi phase and eutectic TiNi + β-Nb phase.All the TiNi-Nb alloys had a dendritic structure. As the Nb content increased, the alloys underwent a densification of the dendritic structure and a grain refinement effect. The eutectic TiNi + β-Nb phase in the TiNi_44_Nb_6_ alloy was located at the grain boundaries and resulted in a significant softening effect.For both deformation regimes (i.e., quasi-static and dynamic), the maximum stress occurred in the TiNi_44_Nb_4_ alloy.As the level of Nb addition increased, the fracture mode transited from a predominantly brittle fracture mode (2 at.% Nb) to a mixed brittle-fracture mode (4 at.%), and finally a predominantly ductile fracture mode (6 at.%).In this study, we determined that the addition of Nb to TiNi alloy could enhance the mechanical properties for engineering applications. In particular, Nb addition can enhance the ductility and strength of the high strain rate application at room temperature.

## Figures and Tables

**Figure 1 materials-15-03124-f001:**
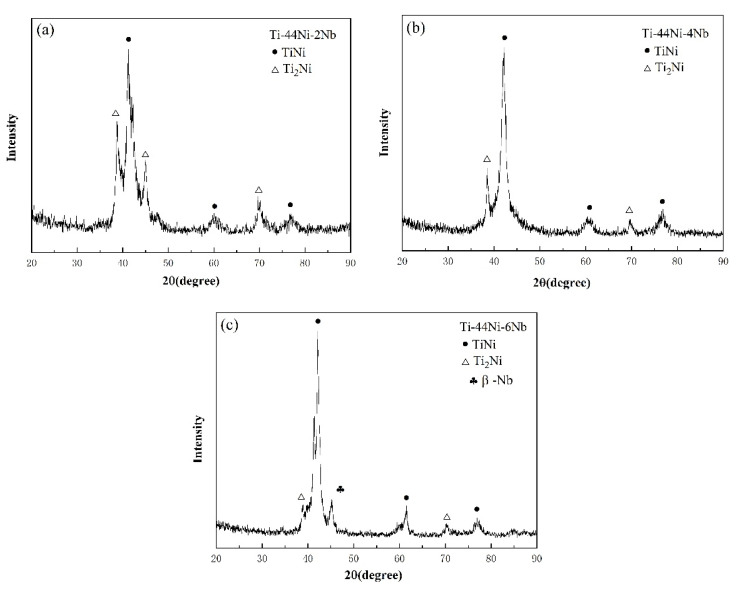
XRD patterns of: (**a**) TiNi_44_Nb_2_, (**b**) TiNi_44_Nb_4_, and (**c**) TiNi_44_Nb_6_ alloys.

**Figure 2 materials-15-03124-f002:**
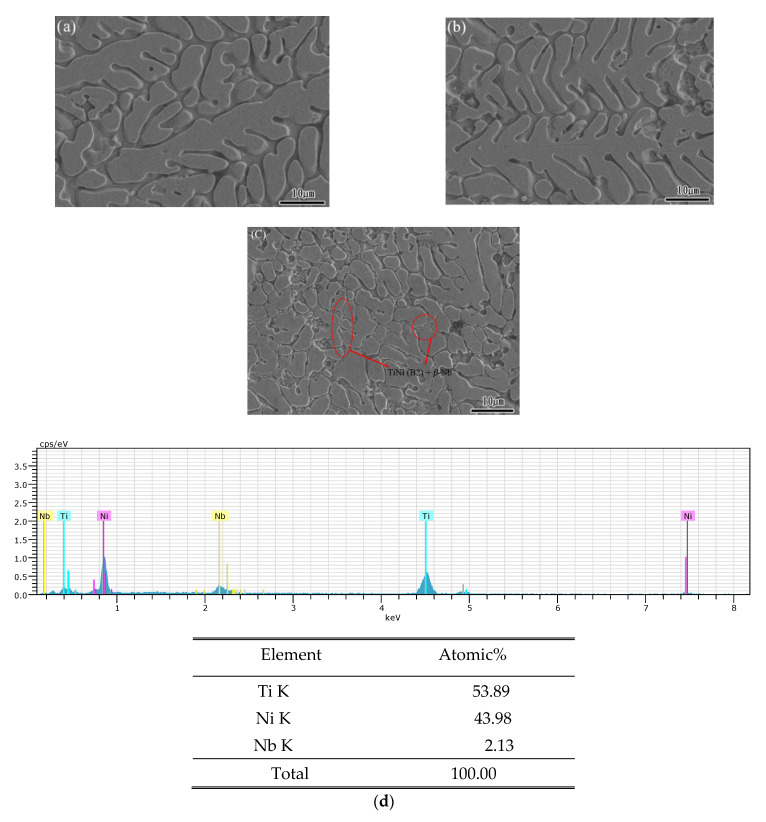
SEM surface morphologies of: (**a**) TiNi_44_Nb_2_, (**b**) TiNi_44_Nb_4_, and (**c**) TiNi_44_Nb_6_ alloys. EDS analysis for (**d**) TiNi_44_Nb_2_, (**e**) TiNi_44_Nb_4_, and (**f**) TiNi_44_Nb_6_ alloys, (**g**) the higher magnification view for (**c**).

**Figure 3 materials-15-03124-f003:**
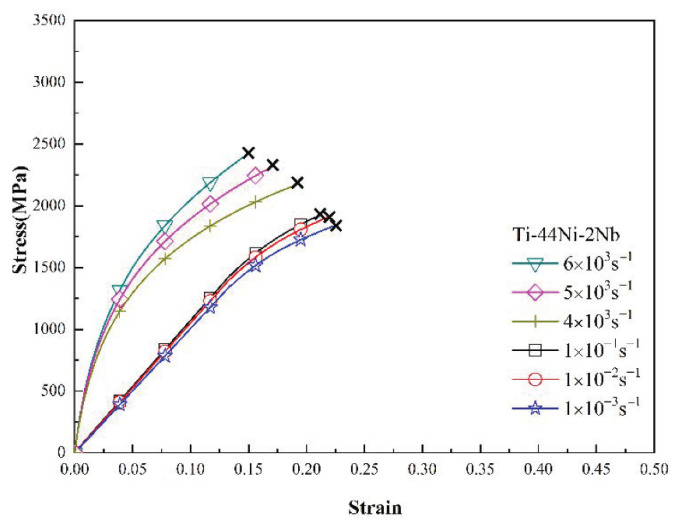
Low strain rate and high strain rate deformation curves of TiNi_44_Nb_2_ alloy.

**Figure 4 materials-15-03124-f004:**
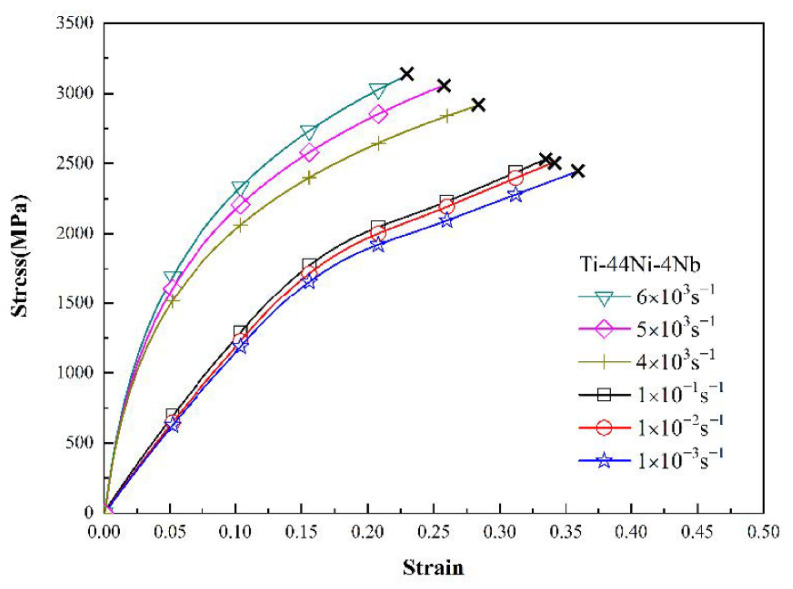
Low strain rate and high strain rate deformation curves of TiNi_44_Nb_4_ alloy.

**Figure 5 materials-15-03124-f005:**
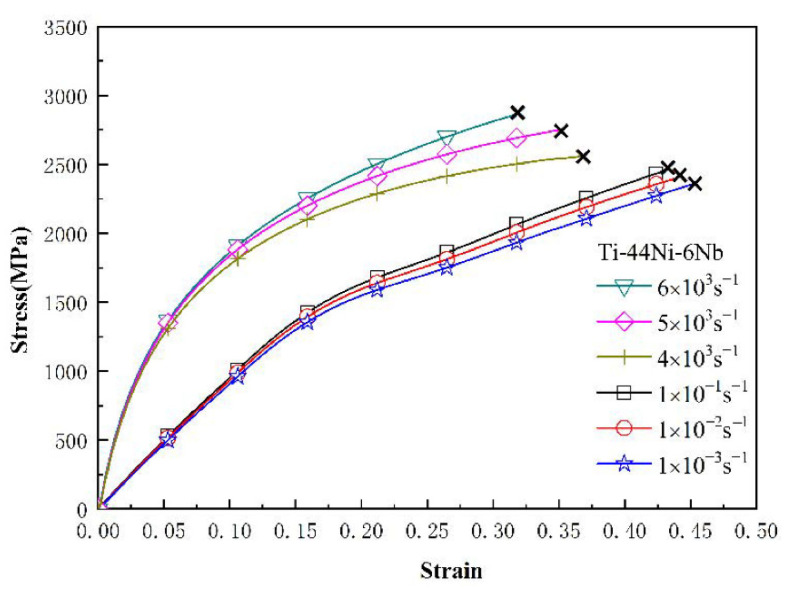
Low strain rate and high strain rate deformation of TiNi_44_Nb_6_ alloy.

**Figure 6 materials-15-03124-f006:**
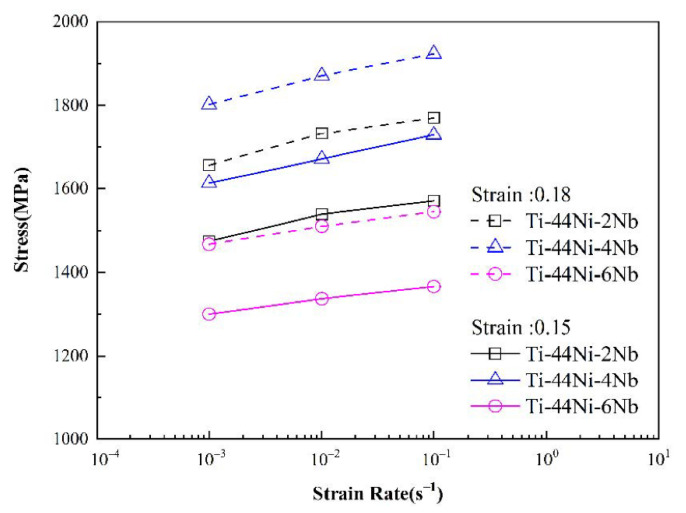
Variation of plastic flow stress with strain rate under static deformation conditions.

**Figure 7 materials-15-03124-f007:**
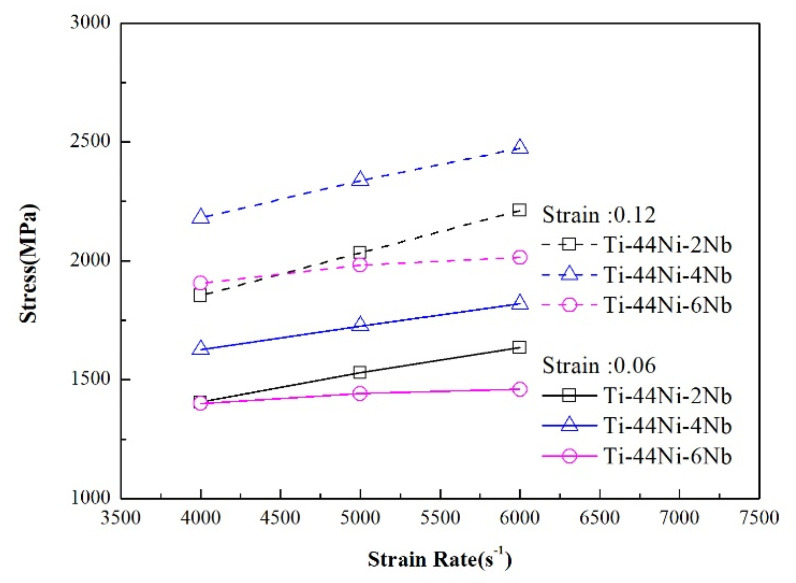
Variation of plastic flow stress with strain rate under dynamic deformation conditions.

**Figure 8 materials-15-03124-f008:**
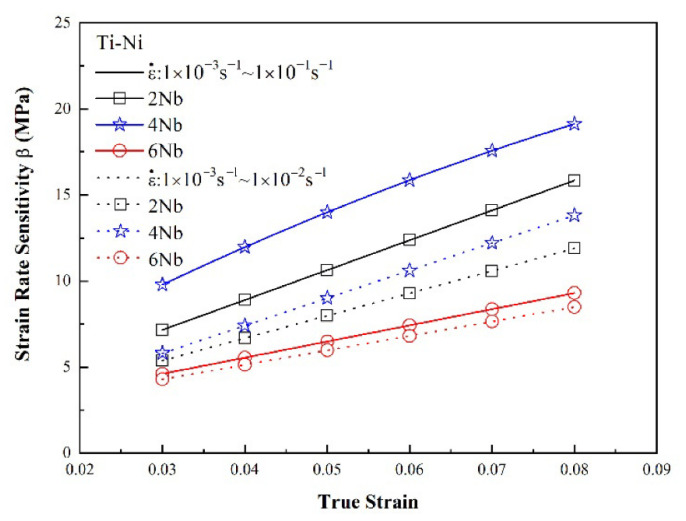
Strain rate sensitivity as a function of true strain under static deformation conditions.

**Figure 9 materials-15-03124-f009:**
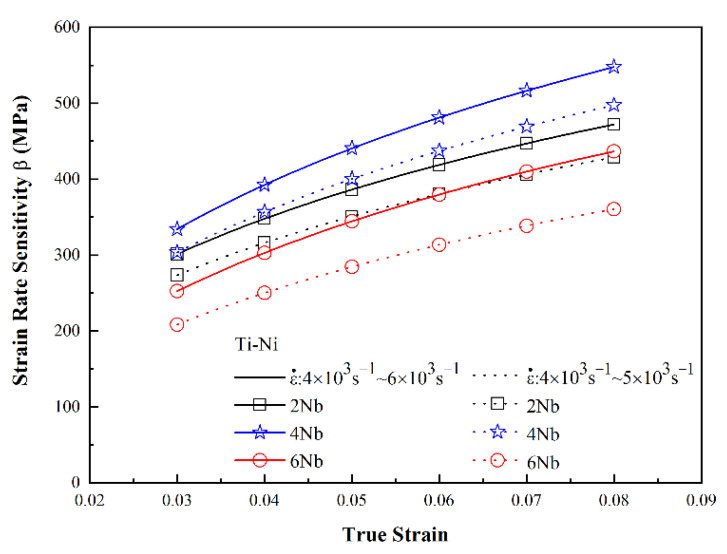
Strain rate sensitivity as a function of true strain under dynamic deformation conditions.

**Figure 10 materials-15-03124-f010:**
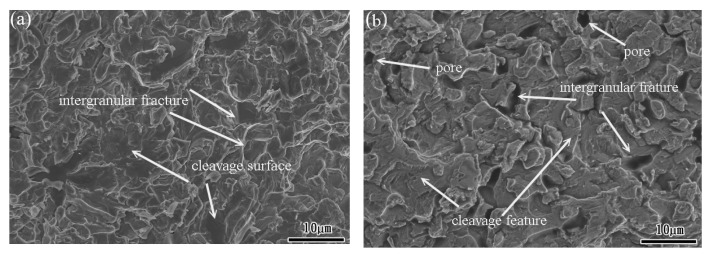
Fracture surface morphology of TiNi_44_Nb_2_ alloy under (**a**) static deformation conditions, and (**b**) dynamic deformation conditions (2000× magnification).

**Figure 11 materials-15-03124-f011:**
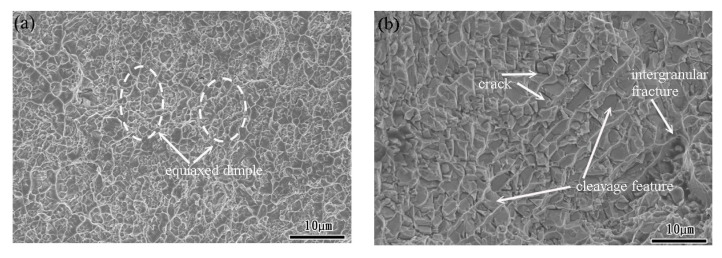
Fracture surface morphology of TiNi_44_Nb_4_ alloy under (**a**) static deformation conditions, and (**b**) dynamic deformation conditions (2000× magnification).

**Figure 12 materials-15-03124-f012:**
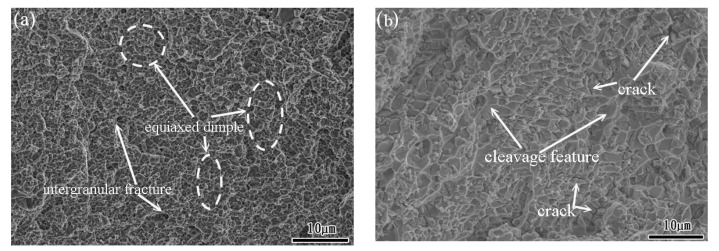
Fracture surface morphology of TiNi_44_Nb_6_ alloy under (**a**) static deformation conditions, and (**b**) dynamic deformation conditions (2000× magnification).

## Data Availability

Not applicable.

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
