# Peer review of "Mechanical Properties and Microstructural Evolution of TiNi-Based Intermetallic Alloy with Nb Addition"

_materials, 2022, doi:10.3390/ma15093124_

Round 1
Reviewer 1 Report
H.F Yang et al investigated the mechanical and microstructural evolution of TiNi based intermetallic alloy with the addition of niobium in different percentages. The intermetallic alloy structure and morphology were characterized by XRD, SEM and optical microscopic techniques. The manuscript is well written and hence I recommend this manuscript should be considered for publication if the author would address the following comments satisfactorily.
- All the figure's resolution needs to be increased and some of the figures are not in the journal format (dimensions).
- The background discussion about the TiNi intermetallic alloys should be included more.
- The characterization of materials in this study looks weak. The author should strengthen the characterization part.
- Why is the reduction in strength of NiTi when increases the doping level is 6% more.
- Why did the author use the OM images instead of SEM for morphology? I don't see any significant changes in the morphology between OM & SEM. SEM images would be recommended for all the cases.
- Why did the author use 2 at % Nb instead of 2% Nb? Any specific reason for using (at %)?
Reviewer 2 Report
- The addition of Nb into the TiNi is a major concept and its effect on the micro and mechanical properties is analyzed. But there is no solid evidence of the Nb effect ad its interdiffusion studies, it should be provided using any characterization techniques.
- The presence of eutectic phases and the interdendritic regions needs to be provided qualitatively.
- Please provide a higher magnification view of the interdendritic regions of the alloys.
- In order to compare the stress-strain curves, it is recommended to use the same scale for X and Y axes.
- In figure 9, what is the difference between dotted lines and solid lines is needs to be indicated clearly.
- The technical discussion of the manuscript should be improved.
- The quality of the figures should be improved.
Reviewer 3 Report
The article ‘Mechanical properties and microstructural evolution of TiNi-based intermetallic alloy with Nb addition, gives an overview of the structure and properties of TiNi alloys with the addition of Nb in several compositions.
The following points are to be addressed by the authors.
- On what basis the percent range of Nb was selected?
- Why only Nb? And no other additives?
- What is the basis of research design and material-process selection? The applications and comparison with published research are missing in the Introduction. The authors should add a concluding summary in the Introduction which should include-
- The gaps in the published research
- Reason (after comparison) to adopt Vacuum Arc Melting Process.
- Reasons to select TiNi-Nb
- Reasons to select the percentage range of Nb
- Was any treatment or preprocessing done before the material/additive processing?
- The article claims to identify the phases in xrd (which is ok) but then mark it on the SEM pictures without any scientific evidence (EDS) that the marked phases are TiNi(B2)+beta-Nb. How can the phases be ascertained to a specific composition without composition analysis at that particular point? If the authors have found similar patterns in other studies, they should cite the reference. Although in such claims only EDAX analysis can be accepted as the patterns of the phases in SEM micrographs can change due to processing method, temperature, superheat degree, etc.
- The EDS studies are needed in the results.
- In Figure 2, apart from the dendritic structure, there are many defects that are visible, particularly in (b) and (c). Please elaborate and discuss.
- In Figure 2, a beautiful dendritic structure is visible, the author may try to measure the size (length) of the primary and secondary dendrites as a function of composition (Nb addition), which will give quite relevant and novel information.
- Figure 10-12, The fractographs should be labeled and the features visible should be marked and annotated.
- There is no information about the scientific outcomes of the research and the applications, a comparative summary highlighting the major novel outcomes and applications of the research should be given at the end of the results and discussion.
- Did the authors try to measure porosity? There is the signature of some pores in the fractographs. Response of the authors is needed on this.
Reviewer 4 Report
Dear authors,
Thank you for your interesting manuscript. I believe it can be accepted for publication after its revision.
I have the following comments:
- Many references do not support the statements given in the article.
1.1 Work [1] on the use of intermetallic compounds as biomaterials for implants, and not "in aerospace, automotive and nuclear energy"
1.2 Work [10] is devoted to the properties of stainless steel grade 316L, and is not related to intermetallic compounds. Only the author Tao-Hsing Chen connects her with the present work.
1.3 Work [11] is devoted to the properties of copper and is not related to the hardness of TiNi alloys and tool steel.
1.4 The work [12] is devoted to the study of the influence of dislocations on the physical and mechanical properties of metals and is not related to the areas of application of alloys based on TiNi.
1.5 Work [14] is devoted to the study of the TiMg alloy and is not related to the areas of application of alloys based on TiNi.
1.6 Works [15, 16] do not consider the effect of adding hafnium (Hf) on the properties of intermetallic compounds.
1.7 Works [17, 18] do not consider the effect of Al addition on grain refinement and increase in hardness.
1.8 Works [19–21] do not contain information about the use of Nb as an alloying element.
1.9 Reference [22] does not investigate the effect of adding Co.
1.10 Work [26] does not use the formula given in the article.
Thus, out of 26 works, only 12 have anything to do with the topic.
- The temperature at which samples are produced 1573K is the same for all samples. Obviously, with an increase in the mass fractions of Nb, the melting point of the intermetallic compound changes. 1573K is more or less than the melting point of the TiNi-Nb alloy. It is known that when the melting temperature is exceeded, less porous samples can be obtained. For the three selected alloys (a) TiNi44Nb2, (b) TiNi44Nb4, and (c) TiNi44Nb6, the same temperature regime actually causes different melting point rises, which causes the formation of different structures shown in Figure 2. Thus, structural differences, and consequently, differences in mechanical properties can be associated with the temperature regime of sample preparation, and not with the % content of Nb.
- It is not clear why argon is not used instead of nitrogen, which provides a higher purity of the vacuum arc melting process.
- There are a number of design notes:
4.1. Used last year's template. Everywhere should be "Materials 2022, 15".
4.2. Words in the title of the article, as well as in sections and subsections, must begin with a capital letter, with the exception of conjunctions, prepositions and articles.
4.3. Some references are not designed according to the template (for example, 3, 22)
4.4. Figures 3–9 are out of proportion.
Round 2
Reviewer 1 Report
The authors have revised their manuscript satisfactorily, hence I recommend this manuscript for publication in the materials journal. However, there are some typo errors (subscripts) in the revised version. The author needs to correct them before publishing online.
Reviewer 2 Report
I appreciated the author's efforts in the revision to enhance its quality.
Reviewer 3 Report
Authors have addressed my comments
Reviewer 4 Report
Dear authors.
Thanks for considering the comments.
I have not other suggestions and recommend this manuscript for publication.